# Florpyrauxifen-Benzyl Selectivity to Rice

**Juan Camilo Velásquez** [1,2,*] , **Angela Das Cas Bundt** [3] , **Edinalvo Rabaioli Camargo** [1] , **André Andres** [4] , **Vívian Ebeling Viana** [1] , **Verónica Hoyos** [5] , **Guido Plaza** [6] and **Luis Antonio de Avila** [1,*]

[1] Crop Protection Graduate Program (Programa de pós Gradução em Fitossanidade), Federal University of Pelotas, Pelotas 96160-000, Brazil; edinalvo.camargo@ufpel.edu.br (E.R.C.); vih.viana@gmail.com (V.E.V.)
[2] Fondo Latinoamericano para Arroz Riego (FLAR), The International Center for Tropical Agriculture (CIAT), Palmira 763537, Colombia
[3] AgriScience, Mogi Mirim 13800-000, Brazil; angela.bundt@corteva.com
[4] Brazilian Agricultural Research Corporation (Embrapa), Cima Temperado Station, Pelotas 96010-971, Brazil; andre.andres@embrapa.br
[5] Facultad de Ingeniería, Universidad del Magdalena, Santa Marta 470004, Colombia; vhoyosc@gmail.com
[6] Departamento de Agronomía, Facultad de Ciencias Agrarias, Universidad Nacional de Colombia, Bogotá D.C. 111321, Colombia; gaplazat@unal.edu.co
* Correspondence: jucvelasquezro03@gmail.com (J.C.V.); luis.avila@ufpel.edu.br (L.A.d.A.)

**Abstract:** Florpyrauxifen-benzyl (FPB) is a new class of auxinic herbicide developed for selective weed control in rice. This study aimed to evaluate the effect of environmental conditions, P450 inhibitors, rice cultivar response, and gene expression on FPB selectivity in rice. Field experiments established in a randomized block design showed that rice plant injury due to two FPB rates (30 and 60 g ai ha$^{-1}$) was affected by planting time and rice stage at herbicide application. The injury was higher at the earliest planting season and more in younger plants (V$_2$) than larger (V$_6$ and R$_0$). However, no yield reduction was detected. Under greenhouse conditions, two dose-response experiments in a randomized block design showed that spraying malathion (1 kg ha$^{-1}$) before FPB application did not reduce herbicide selectivity. The addition of two P450 inhibitors (diethulate and piperonyl butoxide, 10 g a.i. seed-kg$^{-1}$ and 4.2 kg ai ha$^{-1}$, respectively) decreased the doses to cause 50% of plant injury (ED$_{50}$) and growth reduction (GR$_{50}$). However, it seems not to compromise crop selectivity. BRS Pampeira cultivar showed lower ED$_{50}$ and GR$_{50}$ than IRGA 424 RI. A growth chamber experiment was conducted in a completely randomized design to evaluate the gene expression of rice plants sprayed with FPB (30 and 60 g ai ha$^{-1}$). Results showed downregulation of *OsWAKL21.2*, an esterase probably related to bio-activation of FPB-ester. However, no effect was detected on *CYP71A21* monooxygenase and *OsGSTL* transferase, enzymes probably related to FPB degradation. Further research should focus on understanding FBP bio-activation as the selective mechanism.

**Keywords:** herbicide tolerance; metabolism; Rinskor; *Oryza sativa*

## 1. Introduction

The main biotic factor that decreases the yield and quality of rice are weeds (unwanted plants in the field), exhibiting the greatest potential for yield losses globally (34%), which is greater compared to insects (18%) and phytopathogens (15%) [1]. Currently, rice yield loss due to weeds is estimated at around 10% [1]; however, up to 100% losses have been reported in the absence of control [2]. The most important weeds in rice areas include the weedy rice complex (*Oryza sativa* L.), the *Echinochloa* spp. complex (e.g., *Echinochloa crus-galli* (L.) Beauv., *Echinochloa colona* (L.) Link., *Echinochloa oryzicola* Vasinger.), and the *Cyperus* spp. complex (e.g., *Cyperus difformis* L., *Cyperus esculentus* L., *Cyperus iria* L., *Cyperus rotundus* L.) among others such as *Leptochloa* species, *Cynodon dactylon* (L.) Pers., *Eleusine indica* (L.) Gaertn., *Ischaemum rugosum* Salisb., and *Paspalum distichum* L. [3]. These species are difficult to control mainly due to their tolerance to hypoxia and herbicides [3]. Historically, the use of synthetic auxin herbicides (SAHs), such as quinclorac and 2,4-D, allowed for the

post-emergent selective control of weeds in rice [4,5]. However, 17 weed-resistant cases to SAHs in rice crop areas have been reported worldwide, limiting the strategies to prevent rice yield losses [6].

The phenomenon of weed resistance to herbicides is a concern among researchers since the development of this in the field causes an increase in production costs [7]. One example of this is the cost of glyphosate resistance in *Amaranthus palmeri* S. Wats in the United States, which increases the cost of control per hectare up to $40, $54, and $74 in corn, soybeans, and cotton, respectively [7]. In this context, florpyrauxifen-benzyl (FPB), a new class of SAHs (WSSA, HRAC Group 4), has been developed for selective use in rice with a broad spectrum of weed control even in those with confirmed resistance to other herbicides such as quinclorac-resistant *E. crus-galli* [8,9] or glyphosate-resistant *E. colona* [10]. Therefore, FBP has the potential to provide efficient weed management for rice production.

Florpyrauxifen-benzyl, similar to other SAHs, mimics indole-3-acetic acid (IAA), acting as "molecular glue" between the receptor protein complex $SCF^{TIR/AFB}$ (Skp1-cullin-F-box protein) and the co-repressor protein Aux/IAA at the plant cell nucleus, promoting the degradation of Aux/IAA by the ubiquitin-proteosome pathway (26S proteosome) [11]. Aux/IAA inhibits the auxin transcription factors (ARF) associated with the expression of 9-cis-epoxycarotenoid dioxygenase (NCED) in abscisic acid (ABA) biosynthesis and 1-aminocyclopropane-1-carboxylic acid (ACC) synthase in ethylene (ETH) biosynthesis [12,13]. The unbalance of these genes' expression triggers the accumulation of ETH and ABA into plants, which decontrol physiological processes and consequently lead to symptoms and plant death [11].

According to some authors, SAHs have different affinities among $SCF^{TIR/AFB}$ receptors [14,15]. Florpyrauxifen-benzyl has a greater affinity with the $SCF^{TIR/AFB5}$ receptor over other receptors [14,15]. For example, the *Arabidopsis thaliana* L. mutant with a silenced $SCF^{TIR/AFB5}$ was susceptible to 2,4-D but not to FPB [16]. This novelty site of action allows FPB to control quinclorac-resistant *E. crus-galli* accessions [9].

Selectivity in weed management refers to the capacity of a specific herbicide to eliminate weeds in a crop without affecting crop yield or quality [17]. This term may be confused with crop tolerance, which refers to the ability of a plant or population to continue growth or function when the crop is exposed to a potentially harmful agent; thus, both definitions allow us to understand the plant-herbicide interaction [17,18]. This approach is widely used to control weeds in rice crops (e.g., Acetyl-CoA-carboxylase inhibitors: cyhalofop-butyl; photosystem II electron disruptors: propanil), and in many cases, it is highly dependent on the plant's ability to degrade the herbicide [17].

The principal process of how plants dissipate pesticides is metabolic degradation followed by growth dilution and volatilization (76%, 21%, and 3%, respectively) [19]. Cytochrome P450 monooxygenases (CYP450s), glutathione-s-transferase (GST), and glucosyltransferase (GT) are involved in herbicide detoxification [20]. Florpyrauxifen-benzyl is formulated as benzyl ester; thus, the bio-activation is necessary to be toxic (acid form) to plants. Therefore, the activity of an esterase enzyme is required [8,21]. For SAHs-resistant weeds, the esterase bio-activation process has been proposed to occur prior to passing through the cell membrane [22]. The acid form is later metabolized to a hydroxy-acid form, mediated possibly by CYP450 monooxygenation and subsequently conjugating to glucosyl or glutathione transferase enzymes (GT or GST, respectively) [8,23]. The degradation mediated by CYP450 has been reported to confer resistance to other SAHs such as 2,4-D and dicamba in *Papaver rhoeas* L. and *Parthenium hysterophorus* L., but did not affect resistance to picloram [24,25]. Likewise, the lack of esterification of 2,4-DB to 2,4-D permits the selective use of SAHs in a broadleaf crop such as alfalfa (*Medicago sativa* L.) [26].

The selective mechanism of FPB in rice has not been fully described; however, the information reported so far demonstrates that rice tolerance could be related to lack of bio-activation (mediate by lack of esterase activity), metabolic activity (mediated first by CYP450s, and followed by GT or GST), or differences in receptor affinity ($SCF^{TIR/AFB}$) at the

site of action [14,16]. Additionally, FPB has several label recommendations in order to be selective to rice, for example: do no tank mix with malathion (CYP450 inhibitor) or methyl parathion (acetylcholinesterase inhibitor), do not overlap double spray, sensitivity expected of some medium grain and hybrid varieties, do not apply in adverse environmental conditions such as extreme cold or heat, and recommended stages of application in rice at $V_3$–$V_4$ [27].

There are several reports of rice crop response to FPB in field experiments worldwide. For example, in field conditions in Australia, the maximum visual injury reported was 16% using 60 g ai ha$^{-1}$, but this did not affect rice grain yield [28]. Field experiments carried out in Sri Lanka, rice grain yield showed no significant differences with non-treated plots, indicating high levels of selectivity [29]. In Italy, the visual injury was no more than 8% at the BBCH12-21 stage, with no rice grain yield reduction [30]. Field studies in Brazil reported, on average, 4% crop injury ranging from 0 to 30% at a rate of 40 g ai ha$^{-1}$; however, the impact on the grain yield has not been reported for these last studies, and neither has the effect of different planting dates and spraying times [31–34]. Due to the recent introduction of FPB, most of these reports are part of conference proceedings.

Previous rice crop observations have described leaf malformations, stem curling, chlorosis, height, tiller number reduction, and shoot dry weight reduction as common symptomology of FPB in rice [35]. Recent research has investigated rice cultivar's response to FPB regarding the temperature surrounding spraying time, growth stage application, and tank-mix with imazethapyr and malathion [35,36]. However, the effect of planting time, growth, and quantifying the effect of inhibitors on the dose of FPB or genes expression related to metabolism or bio-activation have not been fully described.

Considering the recent introduction of FPB and the variation in the response of rice, it was hypothesized that (I) early planting time and application of FPB at the early rice stage increases rice plant injury and will not result in yield losses; (II) the rice plant injury by FPB will increase by the previous application of P450 inhibitors and would differ across cultivars; and (III) the rapid drop in temperatures after FPB application will reduce the gene expression of candidate genes of CYP450, GST, and esterase. Therefore, the objectives of this research were (I) to evaluate the effect of planting time, plant growth stage at spraying time, and FPB rates on rice crop injury and yield components; (II) to evaluate rice response to FPB doses applied after P450 inhibitor treatment, and to different rice cultivars; and (III) to evaluate the effect of temperature regimes on FPB selectivity and the expression of *CYP71A21*, *OsGSTL3*, and *WALK21.2* (esterase).

## 2. Materials and Methods

### 2.1. Florpyrauxifen-Benzyl Selectivity to Rice, as Affected by Planting Time, Stage, and Rate of Application

2.1.1. Site and Plant Material Description

A field experiment was carried out at the Embrapa Clima Temperado experimental station, Capão de Leão, Rio Grande do Sul (RS), Brazil (31°48′47.76″ S; 52°28′12.28″ W, elevation 18 m above sea level), in 2019/2020 and repeated in the 2020/2021 growing seasons. The soil in the area was sandy-loam, pH 6.3, and organic matter 7 g kg$^{-1}$. Pampa CL cultivar was sowed at a density of 250 plants m$^{-2}$ (90 kg of seed ha$^{-1}$) with a row spacing of 12 cm. This cultivar is tolerant to imidazolinone herbicides and Embrapa recently released it in the south of Brazil. Its' cycle is 113–123 days, its' mean height is 95 cm, and it has long grains with 25.1 g of 1000 grain weight. Soil preparation consisted of two mechanical discs harrowing 15 cm deep four- and two- months before sowing. After the second disc harrowing, a leveling pass was performed using a leveling tool attached to the tractor. The sowing was performed on the dates below described according to the treatments. Fertilization consisted of 350 kg ha$^{-1}$ of NPK (5-20-20) at planting time. Nitrogen was applied 50% before flooding ($V_3$–$V_4$) and 50% at the panicle initiation stage ($R_0$), resulting in the addition of 150 kg ha$^{-1}$. KCl was also applied at 45 kg ha$^{-1}$ at the $R_0$ stage.

### 2.1.2. Experimental Procedures

The experiment was conducted in a randomized block design and two-factorial arrangement with four replications. Factor A included three sowing dates for the 2019/2020 growing season: 30 September, 25 October, and 11 November, and for the 2020/2021 growing season: 25 September, 20 October, and 10 November. These dates correspond to early, medium, and late planting times, respectively. Factor B consisted of three growth stages when herbicide was applied (when 50% of the plants reach: $V_2$: collar formation on leaf two on the main stem; $V_6$: collar formation on leaf six on the main stem; and $R_0$: Panicle development initiated). Factor C comprised three doses of FPB (Loyant®, Corteva Agriscience do Brasil LTDA. At 0, 30, and 60 g ai ha$^{-1}$, corresponding to non-treated check, label rate, and maximum label rate recommended by season). We considered the maximum label rate per season to evaluate the response simulating issues such as overlap double spray or overdoses at tank mix in commercial field conditions. The experimental units consisted of plots that were 2 m wide and 5 m long (10 m$^2$), blocks were separated by 50 cm each other, and treatments into each block were separated by 30 cm.

The weed control program for the experiment consisted of a burndown application with glyphosate (EPSPs inhibitor, ZAPP QI 620®, Monsanto do Brasil LTDA) at 1440 g ae ha$^{-1}$ 30 days prior to planting. A complementary burndown with glyphosate (1440 g ae ha$^{-1}$) plus a pre-emergence herbicide imazapic + imazapyr (ALS inhibitor, Kifix® BASF S.A.) at 24.5 and 73.5 g ai ha$^{-1}$, respectively, was applied when rice was at the spiking stage ($S_3$) [37]. For the 2019/2020 season, imazapic + imazapyr (24.5 and 73.5 g ai ha$^{-1}$) was applied postemergence to keep the plots free of weeds. The addition of acetolactate synthase inhibitor has been reported to be safe to rice when applied in a tank mix or close to FPB application [36]. Florpyrauxifen-benzyl applications were performed for each stage of the application corresponding to each treatment. All herbicide applications were carried out using a backpack sprayer (Herbicat®) with a four-nozzle boom (Tee-Jet AIXR110015) calibrated to deliver 150 L ha$^{-1}$ spraying solution. All applications were performed with wind speed below 4 km h$^{-1}$. Irrigation (flooding) was established in the five days after FPB spraying (7 cm of water sheet averaged over experiment) for $V_2$ treatments, and for $V_6$ and $R_0$ treatments, the fields were partially drained prior to spraying and then re-irrigated after 24 h according to the label recommendations [27].

### 2.1.3. Evaluations

Rice plant injury was evaluated visually where "0%" corresponded to the absence of symptoms and "100%" corresponded to plant death. Plant injury was determined at 3, 7, 14, 21, 28, 35, and 42 days after application (DAA). Rice grain yield was obtained by harvesting panicles in the central area of 3 m$^2$ (1 m width × 3 m length). In addition, by collecting five panicles from each plot, tiller number, grains per panicle, 1000 grain biomass, vain grains per panicle, and unfilled grains per panicle were determined. Based on unfilled grains per panicle divided by grains per panicle, the spikelet sterility percentage was calculated. The grain yield was adjusted to 13% humidity.

### 2.1.4. Statistical Analysis

Normality and homogeneity of variance were analyzed by the Shapiro Wilk test [38], and transformation was not necessary for rice grain yield and yield components (tiller number, grains per panicle, 1000 grain biomass, vain grains per panicle, unfilled grains per panicle, and spikelet sterility percentage). However, due to the high numbers of "0" on plant injury response, it was necessary to convert into a scale of 0.001 to 0.999 proportional to the visual injury scale (0% to 100%) previously described. Variables were analyzed in a mixed model, operating planting time as a random factor using the function *lmer* in the package *lme4*.R, and spraying time and rates were considered as fixed factors [39]. We considered planting time as a random factor considering the lack of repetitiveness of the field conditions. A chi-squared distribution was considered for all response variables. Analysis of variance type III with Satterthwaite's method test was performed to determine



the effect of each factor and its interactions. The analysis of variance showed that there were no significant differences between years (runs of the experiment); thus, the data were pooled. The mean and confidential interval at 95% of eight replicates were used to compare the treatments. All analyses were carried out with the $R^®$ version 3.5.2 GUI 1.70 statistic ambient [40]. We performed a mixed model instead of linear models in order to consider fix and random effects, and this analysis was performed as Oliveira [41] proposed.

### 2.2. Florpyrauxifen-Benzyl Selectivity to Rice, as Affected by P450 Inhibitors, and the Tolerance of Two Cultivars

Two independent experiments were carried out; the first evaluated the addition of P450 inhibitors, and the second evaluated the cultivar response to FPB. Both experiments were carried out in a greenhouse at the Federal University of Pelotas (UFPel), Capão do Leão, RS, Brazil. The first was repeated twice in November 2019 and April 2020, and the second was repeated simultaneously in April 2020. Four rice plants of cultivar IRGA 424 RI and BRS Pampeira were established in 0.5-L pots that were previously filled with rice paddy soil (Albaqualf) collected from the 0–20 cm soil profile from a nearby rice field. Each pot with four plants was considered the experimental unit. IRGA 424 RI plants were used to evaluate the additions of inhibitors, and plants for BRS Pampeira and IRGA 424 RI were used to evaluate cultivars' response. IRGA 424 RI is one of the most used in RS, Brazil, and BRS Pampeira is a recently introduced cultivar in RS by Embrapa.

For both experiments, the inhibitor and the cultivar response experiment, a randomized block design in a two-factor factorial scheme was performed ($8 \times 3$ and $8 \times 2$ factor levels, respectively) with four and three replications, respectively. Eight FPB doses of 0, 15, 30, 60, 120, 240, 480, and 960 g a.i. $ha^{-1}$ were applied for both experiments and considered as "factor A." These doses were considered taking into account the commercial dose (30 g ai $ha^{-1}$). The factor B for the P450 inhibitor experiment consisted of three different levels: inhibitor-non-treated control, malathion (1 kg ai $ha^{-1}$), and dietholate, followed by piperonyl butoxide (PBO) (10 g a.i. seed-$kg^{-1}$ and 4.2 kg ai $ha^{-1}$, respectively). The decision to add two inhibitors (dietholate followed by piperonyl butoxide) was because of a preliminary test that did not show inhibitor effects individually. Dietholate was applied as a seed treatment while malathion and piperonyl butoxide were sprayed one hour before FPB applications. The factor B for the cultivars' response experiment consisted of two cultivars, IRGA 424 RI and BRS Pampeira. The greenhouse temperature was $17 \pm 5\,°C/25 \pm 5\,°C$ (night/day) in November 2019 and $15 \pm 5\,°C/25 \pm 5\,°C$ in April 2020. Relative humidity was $80 \pm 10\%$ averaged over the experiment runs. The estimated photoperiod was 15/9 day/night with 483.5 and 327.9 cal $cm^{-2}$ $day^{-1}$ of solar radiation in November 2019 and April 2020, respectively.

Herbicide applications were applied at the 3-leaf growth stage of rice ($V_3$). A water sheet around five centimeters above the soil ground level was maintained after herbicide application until the end of the experiments. The application of herbicide was carried out with the same equipment and conditions previously described.

Plant injury was evaluated as a visual variable where "0%" corresponded to the absence of symptoms and "100%" corresponded to a dead plant, with symptoms based on chlorosis, wilting, epinasty, leaf malformation, tissue swelling, and stunted growth. Plant injury was determined at 3, 7, 14, 21, and 28 DAA. At 28 DAA, the shoot dry weight was determined by drying the shoot biomass in an air-flow-oven at 60 °C for 48 h. The 50% growth reduction dose values ($GR_{50}$) from shoot dry weight and 50% plant injury ($ED_{50}$) with their corresponding parameters were calculated for each treatment using a logistic model (Equation (1)) function of the *drc* package in $R^®$ version 3.5.2 GUI 1.70 statistic environment [40,42];

$$y = f(x) = C + \frac{D - C}{1 + \exp(b(\log(x) - \log(e)))} \tag{1}$$

where *C* represents the lowest limit, *D* represents the upper limit, *b* describes the slope of curve around the *e* ($ED_{50}$ or $GR_{50}$), the values of *e* corresponded to the rate that reduces to 50% of the response variable *y*, and *x* is the FPB dose in g a.i. $ha^{-1}$.

Florpyrauxifen-benzyl *e* values indices were compared using *EDcomp* function of the *drc* package. The *EDcomp* function compares using the *t*-student test, where $p < 0.05$ indicates significant differences between *e* indices. This same function was operated to compare the curves between runs. Data showed that there were no differences between runs; therefore, the data were pooled. Confidence intervals were calculated using the *confint* function. The $ED_{50}$ of the control without inhibitor and treatments was estimated by an inhibition ratio, as follows: Inhibition ratio = ($ED_{50}$ check without inhibitor—$ED_{50}$ inhibition treatment)/$ED_{50}$ check without inhibitor × 100.

### 2.3. Florpyrauxifen-Benzyl Selectivity to Rice, as Affected by Temperature Regimen and the Expression of Target Candidate Genes

2.3.1. Temperature and Rice Injury

The experiments were carried out in a growth chamber at the Federal University of Pelotas (UFPel), Capão do Leão, RS, Brazil, and it was repeated twice in 2019 and 2020. Nine rice plants of cultivar IRGA 424 RI were established in five-liter pots previously filled with sieved paddy soil (Albaqualf) collected from the 0–20 cm soil profile nearby the rice field.

The experiment was organized in a complete randomized block design in a factorial arrangement with four replications. Factor A consisted of three rates of florpyrauxifen-benzyl: 0, 30, and 60 g a.i. $ha^{-1}$ (30 g a.i. $ha^{-1}$ corresponds to the recommended dose). Factor B corresponded to six temperatures treatments conditions using three regimens and three temperatures. The three temperature regimens considered an initial growing until spraying ($V_3$ rice stage), for 24 h after spraying, and until the end of the experiment (28 days after spraying). The three temperatures were medium (28/25 °C day/night), low (18/15 °C day/night), and high (38/36 °C day/night). The treatments consisted of all-optimum temperatures along with the experiment (T1), medium-low-medium (T2), high-low-medium (T3), medium-high-high (T4), medium-high-medium (T4), and medium-high-low (T5) (Table S1).

The growth chamber was programmed to maintain a controlled day/night temperature according to the treatments previously described under controlled conditions with a photoperiod of 12 h (900 µmol $m^{-2}$ $s^{-1}$) and a constant relative humidity of 70 (±5)%.

Rice plants with three leaves ($V_3$) were applied with FPB. A water sheet around five centimeters above the soil ground level was maintained after herbicide application until the end of the experiments. The application of herbicide was made with the same equipment and conditions previously described in Section 2.1. Visual evaluation of plant injury was assessed for each plant as described in Section 2.1. Plant injury was determined at 3, 7, 14, 21, and 28 DAA. Statistical analyses were performed as described in the field experiment (Section 2.1). There were detected differences between runs at 3, 7, and 28 DAA; however, Levene's test performed to compare the homogeneity of variance over experimental runs found no significance (*p*-value = 0.4051). Therefore, the variance of each experiment was included in the means analysis considering the run as a random factor and estimating the best linear unbiased predictors (BLUPs).

2.3.2. Gene Expression of CYP450, GST, and Esterase Candidate Genes
Plant Material, Growth Conditions, and Treatments

Four commercial rice seeds (*Oryza sativa*. cv IRGA424 RI) were planted at 1 cm depth in each 500 g pot, previously filled with soil as described before. Pots were incubated in a growth chamber at the Federal University of Pelotas, Brazil. A completely randomized design was considered in a factorial scheme with two biological replicates. Four temperature treatments were established (T1, T2, T3, and T4) (Factor A). The initial growth conditions (until application) were 28/25 °C (day/night temperature) for treatment T1, T2, and T4.

For T3, the initial condition was 38/36 °C. Three doses of FPB were sprayed (0, 30, and 60 g ai ha$^{-1}$) when rice reached the three-leaf stage (V$_3$) (Factor B). Twenty-four hours after application time, pots corresponding to each treatment were transferred to specific temperature conditions; T1 continued at 28/25 °C, T2 and T3 decreased to low 18/15 °C, and T4 transferred to high 38/36 °C. Herbicide was sprayed using the same equipment and under the same conditions described above. Approximately eight grams of leaf tissue were collected at each evaluation time (6, 12, and 24 h after spraying) from both biological replicates. Plant material was collected manually from leaf blades using sterilized gloves and scissors. The collected material of each replication was mixed, placed in aluminum foil envelopes, separated by treatment, and immediately after it was collected, it was kept in liquid nitrogen until it was stored in an ultra-freezer −80 °C. The whole material was frozen with liquid nitrogen and macerated using laboratory mortar and pestle, and then the material was stored in microtubes of 1.5 mL at −80 °C.

RNA Extraction and cDNA Synthesis

Total RNA was extracted from two grams of leaf tissue using PureLink Plant RNA Reagent (Invitrogen®) following the protocol described by the manufacturer. The quantity and purity of the RNA was verified by spectrophotometry in Nanovue (GE Healthcare) and integrity by agarose gel electrophoresis. Each RNA sample (1 μg) was treated with DNase I Amplification Grade (Invitrogen®) and was converted into cDNA using oligo(dT) and the SuperScript III first-strand system kit (Invitrogen®).

Gene Expression Quantification by RT-qPCR

The RT-qPCR experiment was performed following the MIQE guidelines [43] using oligonucleotides for the reference and target genes (Table S2). Validation experiments were performed using four cDNA dilutions in order to determine the amplification efficiency and specificity of each oligonucleotide. Those that were 90–110% efficient and with only one peak in the dissociation curve were used.

Gene expression assay was performed with the LightCycler® 480 Instrument II (Roche) thermocycler using three biological and three technical replicates. Reactions were performed using cDNA 1 μL in 1:25 dilution (determined during validation experiments), UltraPure™ DNase/RNase-Free Distilled Water (Invitrogen) 11.0 μL, ROX Reference Dye (Invitrogen) 0.25 μL, 10X PCR Buffer 2.0 μL, MgCl 50 mM 1.5 μL, Platinum™ Taq DNA Polymerase (Invitrogen) 0.05 μL, dNTPs 0.2 μL, SYBR Green I (Invitrogen) 3.0 μL, and oligonucleotide 10 mM 1.5 μL for each forward and reverse in 20 μL of the final volume reaction. Negative control reactions without cDNA were also performed for each oligonucleotide pair. The PCR condition was of initial denaturation at 95 °C for 5 min, 45 cycles of 95 °C for 20 s, 60 °C for 15 s, and 72 °C for 20 s. Reactions were performed in LightCycler® 480 Multiwell Plates 96 (Roche).

Target gene expression quantification was conducted using the ΔΔCT method [44] using, as the baseline, the expression of the T1 treatment (28/25 °C along with the experiment) without herbicide application normalized with respect to *Os18S*, *OsEF1α*, and *OsUBQ5* reference genes [45]. Gene expression data were converted in Log$_2$-fold change. The mean and confidence interval at 95% of three technical replicates were used to compare treatments.

## 3. Results

### 3.1. Florpyrauxifen-Benzyl Selectivity to Rice, as Affected by Planting Time, Stage, and Rate of Application

In general, there was a significant *p*-value in at least one of the double interactions for plant injury throughout the evaluation time (Table 1). The double interaction of spraying time and rates were significant at all evaluations times. Year operated as a fixed factor and showed non-significant *p*-values among evaluation times. Thus, this supports the decision pool data over the years for posterior analyses.

Maximum plant injuries were observed at high doses (60 g ai ha$^{-1}$), sprayed at V$_6$ for early planting time 7 DAA (27.4%), sprayed at V$_2$ for late planting time 14 DAA (34.8%), and sprayed at V$_2$ for medium planting time 21 DAA (34.4%) (Figure 1). Regarding the planting timing, injuries in late planting time (November) showed lower values than early (September) and medium (October) planting time. Moreover, at late planting time, plants seem to have a better recovery from the injury than early or medium planting time. Herbicide applications at R$_0$ did not show differences in plant injury compared to the non-treated in neither planting time nor FPB rate. In contrast, applications made at V$_2$ spraying time showed more significant plant injuries than V$_6$, excluding at 7 DAA, where there were more injuries at V$_6$ than at V$_2$. Mostly, injuries were higher at double the label rate (60 g ai ha$^{-1}$) than at the label rate (30 g ai ha$^{-1}$). Likewise, plant injuries at 42 DAA for label rates were greater at V$_2$ than at V$_6$, suggesting better plant recovery for applications at V$_6$.

The environmental observations at the earliest applications suggested that low radiation and low temperature surrounding the application time were related to increased injuries. Solar radiation averaged over five days before treatment was lower for early planting time (665.6 and 702.8 µmol m$^{-2}$ s$^{-1}$ for 2019 and 2020, respectively), compared to medium (1401.9 and 946.1 µmol m$^{-2}$ s$^{-1}$ for 2019 and 2020, respectively) and late planting time (1574.2 and 1306.1 µmol m$^{-2}$ s$^{-1}$ for 2019 and 2020, respectively) (Figure S1). Likewise, solar radiation was lower over the five days after V$_2$ treatments at early (1126.6 and 1179.5 µmol m$^{-2}$ s$^{-1}$ for 2019 and 2020, respectively) than medium (1206.7 and 1306.1 µmol m$^{-2}$ s$^{-1}$ for 2019 and 2020, respectively) and late planting time (1359.2 and 1048,7 µmol m$^{-2}$ s$^{-1}$ for 2019 and 2020, respectively) (Figure S1).

The injuries observed in this study did not cause lasting effects and did not affect rice yield. Thus, the tiller number, the number of grains per panicle, the number of unfilled grains per panicle, grain yield, and sterility were not affected by any treatments (Table S3). The applications made at the early reproductive stage (R$_0$) did not show more than 3.8% of injury for all planting times, and there was no impact on grain yield or yield components (Figure 1 and Table S3).

**Table 1.** The *p*-values of type III analysis of variance with Satterthwaite's method for rice plant injury applied with three rates of florpyrauxifen-benzyl through time after treatment and as affected by three planting times, three spraying times, and their interactions.

| Factor | Pr (>F) | | | | | |
|---|---|---|---|---|---|---|
| | **7 DAA** [1] | **14 DAA** | **21 DAA** | **28 DAA** | **35 DAA** | **42 DAA** |
| Year | 0.1325 | 0.2345 | $4 \times 10^{-11}$ | 0.2744 | 0.0953 | 0.0928 |
| Planting time | $1 \times 10^{-11}$ | 0.070 | 0.044 | 0.010 | 0.002 | 0.004 |
| Spraying time | $2 \times 10^{-16}$ | $2 \times 10^{-16}$ | $2 \times 10^{-16}$ | $2 \times 10^{-13}$ | $3 \times 10^{-12}$ | $1 \times 10^{-10}$ |
| Rate | $2 \times 10^{-15}$ | $2 \times 10^{-16}$ | $2 \times 10^{-16}$ | $1 \times 10^{-15}$ | $4 \times 10^{-12}$ | $1 \times 10^{-10}$ |
| Planting time × Spraying time | $2 \times 10^{-12}$ | $3 \times 10^{-4}$ | 0.028 | 0.080 | 0.225 | 0.194 |
| Planting time × Rate | $1 \times 10^{-5}$ | 0.393 | 0.267 | 0.022 | 0.039 | 0.169 |
| Spraying time × Rate | $3 \times 10^{-10}$ | $5 \times 10^{-12}$ | $8 \times 10^{-15}$ | $2 \times 10^{-7}$ | $3 \times 10^{-6}$ | $9 \times 10^{-6}$ |
| Planting time × Spraying time × Rate | $7 \times 10^{-6}$ | 0.058 | 0.027 | 0.238 | 0.338 | 0.539 |

[1] Abbreviations: DDA, days after application.

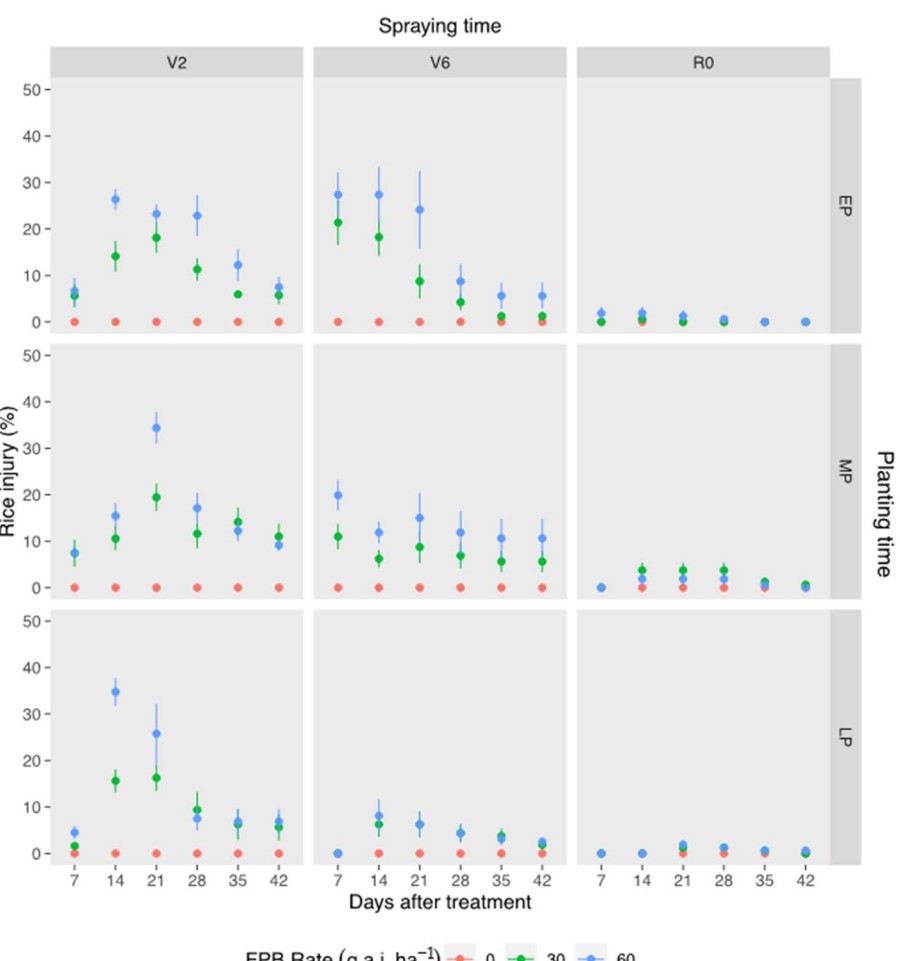

**Figure 1.** Rice crop injury as affected by planting time and florpyrauxifen-benzyl (FPB) application time and rate. Average of two growing seasons (2019/20 and 2020/21). EP: early planting time (30 September 2019/25 September 2020), MP: medium planting time (25 October 2019/20 October 2020), and LP: late planting time (11 November 2019/10 November 2020). Growth stage at spraying time, when 50% of crop reached $V_2$: collar formation on leaf two on the main stem; $V_6$: collar formation on leaf six on the main stem; $R_0$: Panicle development has initiated. Confidence interval at 95% (*n* = 8).

*3.2. Florpyrauxifen-Benzyl Selectivity to Rice, as Affected by P450 Inhibitors, and the Tolerance of Two Cultivars*

3.2.1. Effect of P450 Inhibitors on Rice Response to Florpyrauxifen-Benzyl

Based on the confidence interval of $GR_{50}$ calculated from the dry shoot weight collected at 28 DAA, there was a significant difference between the check without inhibitor and with dietholate followed by PBO inhibitor, whereas for the treatment with malathion, it did not show a difference (Figure 2). The dietholate followed by PBO showed the lowest $GR_{50}$ with a 63% inhibition ratio compared to FPB without the inhibitor; however, as observed with $ED_{50}$, the dose to reach $GR_{50}$ was 2.9-fold times the recommended label rate (89.6 g ai ha$^{-1}$) (Table 2).

Generally, the efficient doses of FPB for rice plant injury ($ED_{50}$) through time were numerically lesser when plants were treated with inhibitors than plants without inhibitors (Table S4). Likewise, the dietholate followed by PBO treatments showed lower $ED_{50}$ in comparison with both the check without inhibitor and with malathion (Table S4). However, the *p*-value of the *t*-student test only detected significant differences for $ED_{50}$ at 21 DAA between the check without inhibitor and dietholate followed by PBO treatment. Although, at 28 DAA, there was no detected significance, this treatment reaches almost 40% of inhibition with a *p*-value = 0.06.

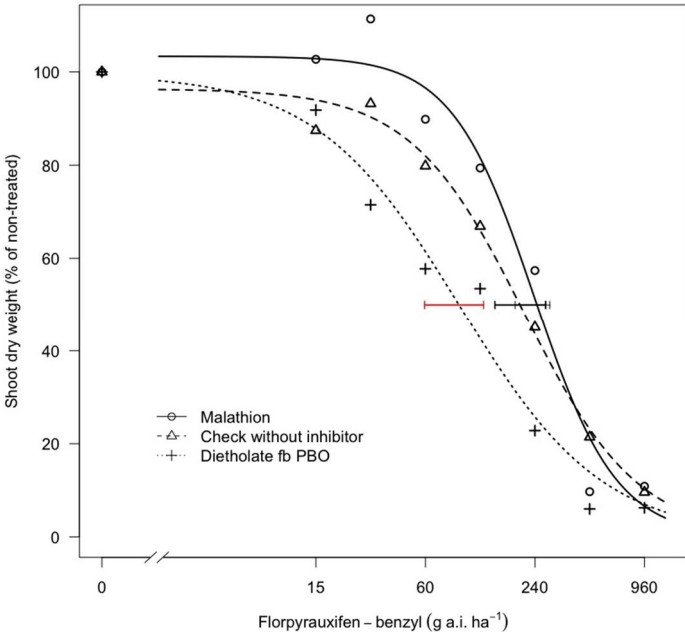

**Figure 2.** Modeled dose-response of rice growth reduction sprayed with florpyrauxifen-benzyl and applied with two P450 inhibitor treatments (malathion (1000 g ai ha$^{-1}$) and dietholate (10 g ai seed-Kg$^{-1}$) followed by (fb) piperonyl butoxide (PBO) (4200 g ai ha$^{-1}$)). Dietholate was applied as a seed treatment, while malathion and piperonyl butoxide were applied one hour before florpyrauxifen-benzyl spraying. Confidence interval is estimated at the 50% of growth reduction, using the *confint* function in the drc R-package. Mean over experimental runs (*n* = 8).

**Table 2.** Parameters estimate of the dose response curve of rice dry shoot weight evaluated at 28 days after florpyrauxifen-benzyl treatment as affected by P450 inhibitors applied one hour before for malathion and piperonyl butoxide (PBO), and at seed treatment for dietholate.

| Treatments | B [1] | SE [2] | Dry Shoot Weight (g plant$^{-1}$) | | GR$_{50}$ [4] | CI [5] 95% | Inhibition Ratio (%) [6] | *p*-Value [7] |
|---|---|---|---|---|---|---|---|---|
| | | | D [3] | SE | g a.i. ha$^{-1}$ | | | |
| Check without inhibitor | 1.4 | (0.3) | 4.4 | (0.2) | 216.9 | (144.5–274.7) | | |
| Malathion | 2.0 | (0.4) | 4.0 | (0.2) | 243.0 | (186.9–289.6) | 0.0 | 0.573 |
| Dietholate fb PBO | 1.1 | (0.2) | 3.5 | (0.2) | 89.6 | (59.5–125.4) | 63.1 | 0.031 |

[1] Slope around GR$_{50}$. [2] SE: standard error. [3] Upper limits for all plants. [4] Doses of florpyrauxifen-benzyl (g a.i. ha$^{-1}$) causes 50% of grow reduction. [5] CI: confidence interval. [6] (GR$_{50}$ check without inhibitor—GR$_{50}$ inhibition treatment)/GR$_{50}$ check without inhibitor × 100. [7] Florpyrauxifen-benzyl vs. inhibition treatment fb florpyrauxifen-benzyl on rice dry shoot weight *t*-statistics comparison of GR$_{50}$. *p*-value > 0.05 means non-significant difference between treatments.

### 3.2.2. Rice Cultivar Response to Florpyrauxifen-Benzyl Application

The response of both IRGA 424 RI and BRS Pampeira to FPB doses at 28 DAA are observed in Figure 3. The confidential intervals at ED$_{50}$ and GR$_{50}$ do not intersect between cultivars, suggesting a significant difference (Figure 3). The rice cultivar BRS Pampeira was more sensitive than IRGA 424 RI to the dose rising of FPB in both plant injuries and shoot dry weight reduction.

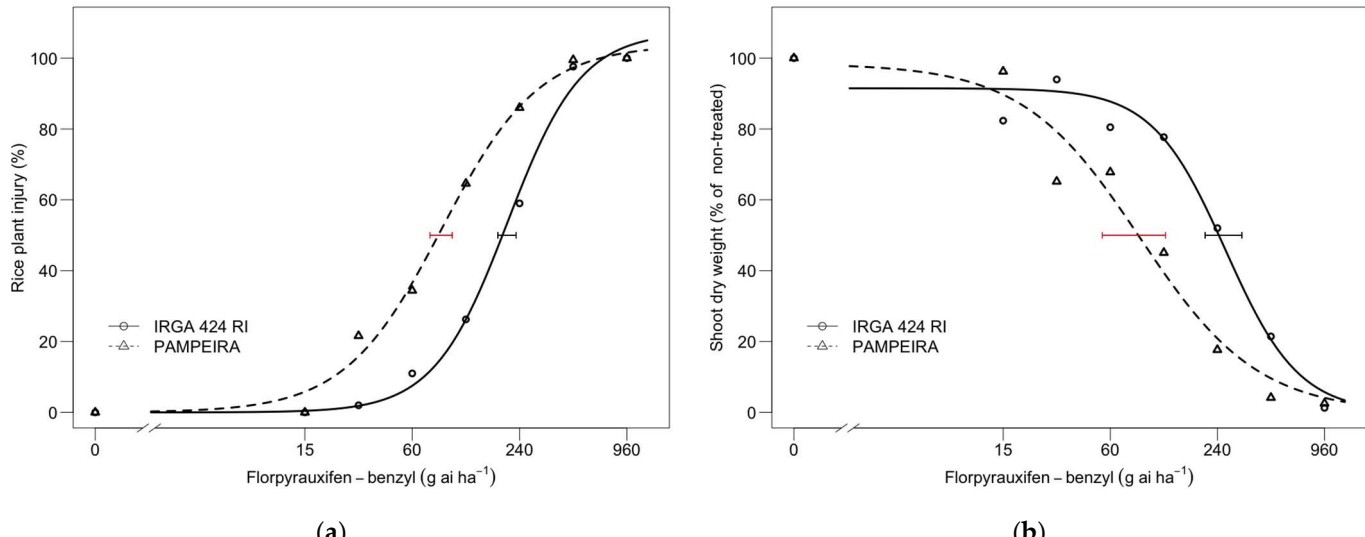

**Figure 3.** Rice cultivar BRS Pampeira and IRGA 424 RI response to florpyrauxifen-benzyl doses to (**a**) plant injury and (**b**) shoot dry weight relative to the non-treated check. The confidential interval was estimated at 50% of plant injury and growth reduction, respectively. Means over experimental runs (*n* = 6).

The $ED_{50}$ and $GR_{50}$ values showed that BRS Pampeira was 2.3-fold and 3.0-fold times lesser than the IRGA 424 RI, respectively (Table 3). *T*-student tests were used to analyze the differences between cultivars for the $ED_{50}$ and $GR_{50}$ values; as observed at the *p*-value, there was a significant differential response of rice cultivars to FPB being less tolerant of BRS Pampeira than IRGA 424 RI (Table 3).

**Table 3.** Parameters estimate of the dose-response curve of rice plant injury and rice dry shoot weight evaluated at 28 days after florpyrauxifen-benzyl treatment to two rice cultivars.

| Cultivar | B [1] | SE [2] | Rice Plant Injury (%) | | $ED_{50}$ [4] | CI 95% [5] | *p*-Value [6] |
|---|---|---|---|---|---|---|---|
| | | | D [3] | SE | g a.i. ha$^{-1}$ | | |
| IRGA 424 RI | −2.1 | (0.2) | 100 | (3.7) | 205.5 | (181.2–229.8) | |
| BRS Pampeira | −1.6 | (0.1) | 100 | (3.2) | 88.0 | (75.4–100.5) | 0.000 |
| | | | Dry shoot weight (g) | | $GR_{50}$ [5] | CI 95% | |
| IRGA 424 RI | 2.1 | (0.5) | 4.6 | (0.2) | 267.2 | (204.9–329.4) | |
| BRS Pampeira | 1.3 | (0.2) | 4.5 | (0.3) | 88.3 | (54.2–122.5) | 0.004 |

[1] Slope around $ED_{50}$ and $GR_{50}$. [2] SE: standard error. [3] Upper limit for all plants. [4] Doses of florpyrauxifen-benzyl (g a.i. ha$^{-1}$) cause 50% of plant injury and growth reduction. [5] Confidence interval. [6] IRGA 424 RI vs. BRS PAMPEIRA on plant injury and rice dry shoot weight *t*-statistics comparison of $ED_{50}$ and $GR_{50}$. *p*-value > 0.05 means non-significant difference between treatments.

### 3.3. Florpyrauxifen-Benzyl Selectivity to Rice, as Affected by Temperature Regimen and the Expression of Target Candidate Genes

#### 3.3.1. Temperature and Rice Injury

There were detected differences over runs for plant injury at 3, 7, and 28 DAA (Table 4); therefore, this effect was analyzed as a random factor, and we calculated the corresponding BLUPs to predict the effect of runs (Table S5). Generally, injuries through evaluations were lesser for treatments where temperatures keep all medium or decrease until low for 24 h after spraying (T1, T2, and T3) than those where temperature increased from medium to high for 24 h after spraying (T4, T5, T6) (Table 4). There was an exception for T5 at 7 and 14 DAA, where injuries were not different to the non-treated and similar to treatments where temperature decreased from medium to low for 24 h after spraying. However, at 21 and 28 DAA, injuries were similar to those where temperature increased from medium

to high 24 h after application (Table 4). Thereby, results in this research suggest that the increase in air temperature 24 h after spraying (from 28/25 °C to 38/36 °C) promoted the rice injury to FPB.

**Table 4.** Effect of six temperature treatments and three florpyrauxifen-benzyl rates on rice plant injury at 3, 7, 14, 21, and 28 days after application.

| Treatment [1] | Rate [2] | Rice Plant Injury (%) | | | | | | | | | |
|---|---|---|---|---|---|---|---|---|---|---|---|
| | | 3 DAA [3] | | 7 DAA | | 14 DAA | | 21 DAA | | 28 DAA | |
| Non-treated | 0 | 0 | h | 0 | h | 0 | e | 0 | d | 0 | f |
| T1 (all medium) | 30 | 3.3 | fg | 5.5 | efg | 2.8 | de | 4.0 | c | 1.8 | f |
| | 60 | 6.5 | cde | 11.3 | b | 11.0 | bc | 10.4 | b | 2.3 | f |
| T2 (med-low-med) | 30 | 3.5 | f | 3.9 | fg | 6.1 | cde | 2.6 | cd | 1.9 | f |
| | 60 | 4.0 | ef | 7.2 | cdef | 5.3 | cde | 1.7 | cd | 1.7 | f |
| T3 (high-low-med) | 30 | 2.4 | fgh | 4.0 | efg | 8.1 | cd | 4.6 | c | 3.3 | ef |
| | 60 | 5.3 | def | 6.6 | defg | 10.6 | bc | 4.3 | c | 2.5 | f |
| T4 (med-high-high) | 30 | 8.9 | bc | 7.6 | cde | 16.5 | b | 10.0 | b | 6.9 | de |
| | 60 | 14.4 | a | 20.5 | a | 24.2 | a | 12.4 | ab | 11.9 | b |
| T5 (med-high-med) | 30 | 8.2 | bcd | 3.2 | gh | 8.0 | cd | 10.8 | b | 7.8 | cd |
| | 60 | 14.5 | a | 4.7 | efg | 5.6 | cde | 11.1 | b | 10.9 | bc |
| T6 (med-high-low) | 30 | 7.8 | bcd | 10.8 | bc | 11.3 | bc | 13.5 | ab | 10.9 | bc |
| | 60 | 10.0 | b | 9.4 | bcd | 16.4 | b | 14.6 | a | 21.9 | a |
| | | Pr (>F) | | | | | | | | | |
| Rate | | $2.20 \times 10^{-16}$ | | $2.20 \times 10^{-16}$ | | $2.20 \times 10^{-16}$ | | $2.20 \times 10^{-16}$ | | $2.20 \times 10^{-16}$ | |
| Temperature treatment | | $2.20 \times 10^{-16}$ | | $3.84 \times 10^{-8}$ | | $1.05 \times 10^{-8}$ | | $2.20 \times 10^{-16}$ | | $2.20 \times 10^{-16}$ | |
| Run | | $2.20 \times 10^{-16}$ | | 0.0253 | | 0.7306 | | 0.4464 | | $1.23 \times 10^{-5}$ | |
| Rate: Temperature treatment | | $5.62 \times 10^{-9}$ | | $3.83 \times 10^{-8}$ | | 0.000121 | | $3.50 \times 10^{-12}$ | | $2.20 \times 10^{-16}$ | |

[1] See Table S3 for temperature treatments. [2] Florpyrauxifen-benzyl rate in g ai ha$^{-1}$. [3] Abbreviations: DAA, days after application. Mean aggrupation with different letters means significant differences for fixed effects into each planting time by the Kenward-roger method and across random factor (PT) when necessary (confidence level 95%).

### 3.3.2. Cytochrome P450 Monooxygenase Expression in Rice as Affected by Florpyrauxifen-Benzyl and Temperature

Six hours after the spraying (HAS) at commercial and double doses, *CYP71A21* expression was lesser than those non-treated for rice growth at 28/25 °C, whereas it was greater than those non-treated for rice growth at 38/35 °C, but it was not consistent at the double dose (Table 5). At 12 HAS, the T1 treatment, which continues at medium temperature (28/25 °C), showed greater expression of *CYP71A21* after spraying FPB, of 0.6-, 1.6-, and 1.8-fold, for non-treated, 30, and 60 g ai ha$^{-1}$, respectively. However, it was not consistent with 12 and 24 HAS, and contrastingly, all *CYP71A21* gene expression was downregulated even in the non-treated check treatment.

**Table 5.** Relative mRNA abundance (Log$_2$-fold change of gene expression) in rice (*Oryza sativa* cv IRGA 424 RI) leaves after florpyrauxifen-benzyl application. mRNA abundance of each gene from non-treated plants served as the baseline for determining relative RNA levels. The color scale below the heatmap shows the expression level, and values in parentheses represent the confidence interval (*n* = 4).

| Florpyrauxifen-Benzyl | | Log$_2$-Fold Change of Gene Expression | | | | | | | | | |
| --- | --- | --- | --- | --- | --- | --- | --- | --- | --- | --- | --- |
| | | T1 All Medium | | | T2 Med-Low | | T3 High-Low | | | T4 Med-High | |
| Gene | Doses | 6 HAS [1] | 12 HAS | 24 HAS | 12 HAS | 24 HAS | 6 HAS | 12 HAS | 24 HAS | 12 HAS | 24 HAS |
| CYP71A21 | 0 | 3.4 (0.32) | −0.6 (0.02) | −3.1 (0.17) | 1.9 (1.06) | −1.9 (0.14) | −0.8 (0.35) | −4.3 (0.81) | −4.4 (1.66) | −3.4 (1.32) | −3.0 (0.46) |
| | 30 | −0.9 (0.07) | 1.6 (0.21) | −2.1 (0.84) | −0.1 (0.08) | −4.0 (0.63) | 2.5 (0.25) | −7.6 (1.82) | −5.1 (1.22) | −4.6 (1.94) | −2.7 (0.91) |
| | 60 | 0.7 (0.13) | 1.8 (0.55) | −4.3 (1.00) | −4.9 (1.32) | −3.0 (1.28) | 0.6 (0.83) | −5.0 (1.12) | −3.4 (0.64) | −3.3 (0.66) | −4.6 (0.81) |
| OsGSTL3 | 0 | 0.20 (0.37) | 0.3 (1.15) | −0.8 (0.44) | 0.6 (0.02) | −2.2 (0.32) | −2.3 (0.87) | −0.5 (0.68) | −3.3 (0.33) | −1.0 (0.67) | −6.1 (1.91) |
| | 30 | −0.7 (0.18) | 1.1 (0.96) | −2.4 (1.31) | 0.4 (0.50) | −3.3 (1.12) | 0.9 (0.00) | −4.0 (1.35) | −3.6 (1.79) | −1.6 (0.30) | −5.4 (1.65) |
| | 60 | 2.9 (1.01) | 0.2 (0.18) | −3.0 (0.66) | −2.7 (0.59) | −2.7 (1.13) | −0.9 (0.16) | −2.0 (0.88) | −3.5 (0.85) | −1.5 (0.62) | −6.2 (1.37) |
| WAKL21.2 | 0 | 6.9 (0.16) | 6.6 (0.81) | 4.9 (0.04) | 3.1 (0.10) | 4.5 (0.20) | 5.0 (0.41) | 4.6 (0.48) | −1.0 (0.98) | 5.7 (0.92) | 1.7 (0.73) |
| | 30 | 3.4 (0.28) | 5.4 (0.89) | 2.7 (0.70) | 5.6 (1.16) | 0.5 (0.01) | 0.1 (0.00) | −1.2 (0.09) | −2.0 (0.51) | 3.5 (0.77) | 2.6 (0.34) |
| | 60 | 2.9 (0.90) | 5.4 (0.39) | 3.3 (0.78) | 5.0 (0.32) | 2.4 (0.65) | 1.7 (0.64) | 0.0 (0.56) | 3.0 (0.36) | 3.6 (1.20) | 0.6 (0.33) |
| Scale | | | | | | | | | | | |
| | Expression scale (Log$_2$-fold change) | | | | <−8 | −8 to −4 | −4 to −2 | −2 to 0 | 0 to 2 | 2 to 4 | >4 |

[1] HAS = hours after herbicide spraying florpyrauxifen-benzyl; T1, T2, and T4 share same conditions at initial growing (until 6 HAS); thus, the same expression was considered for them.

### 3.3.3. Glutathione S-Transferase Expression in Rice as Affected by Florpyrauxifen-Benzyl and Temperature

Six hours after FPB spraying, an overexpression for double doses was observed, whereas the high to low treatment (T3) showed a slight upregulation at commercial doses (Table 4). At 12 HAS, an increase in expression of *OsGSTL3* was observed for treatments at constant temperature (T1: all medium, 28/25 °C) at commercial doses compared to non-treated. Treatment where temperature change from medium to low (T2: 28/25 °C to 18/15 °C) showed downregulation at double of the commercial dose. The rest of the treatments at 12 and 24 HAS showen downregulation even in the non-treated check treatments.

### 3.3.4. OsWAKL21.2 Expression in Rice as Affected by Florpyrauxifen-Benzyl and Temperature

Generally, the non-treated plants showed positive values of *OsWAKL21.2* for all temperature treatments at 6, and 12 HAS (Table 4). However, at 24 HAS, a decrease was observed in expression for non-treated treatment for temperature changes from high to low and medium to high (T3: 38/35 °C to 18/15 °C, and T4: 28/25 °C to 38/35 °C, respectively) compared to all medium temperature and medium to low (T1: 28/25 °C and T2: 28/25 °C to 18/15 °C, respectively) (Table 4), whereas T3 at 60 g a.i. ha$^{-1}$ and T4 at 30 g a.i. ha$^{-1}$ were upregulated 3-fold and 2.5-fold times compared to the non-treated. Therefore, generally, the addition of FPB suggests a decrease in the expression of *OsWAKL21.2* in rice, but an increase in temperatures before or after FPB spraying led to upregulation, which could be related to increasing the bio-activation of FPB-ester.

### 4. Discussion

Our results were similar to others experiments carried out in greenhouse conditions in Arkansas, where the rice injury caused by FPB (30 g ai ha$^{-1}$) was higher at 1-leaf (15%) than at 5-leaf rice (4%) [35]. Raising the rate of FPB and the growth stage of spraying was similar to the results observed for triclopyr (another auxin herbicide). The increased triclopyr rate raised the rice crop injury in different cultivars, being higher when sprayed at 0.8 than 0.6 Kg ai ha$^{-1}$, and exposed greater injury when sprayed at the early stage of growth (V$_2$: 65–45% V$_4$: 40–15%) than at late stages (panicle initiation: 0%) [46]. Those studies

discuss that smaller plants can be more susceptible to herbicide since more concentration of herbicide may reach the plant tissue. Likewise, larger rice plants are able to metabolize and detoxify auxin herbicides. However, for instance, at 7 DAA, we observed more injuries at $V_6$ applications than $V_2$, and these results also were observed in a medium-grain rice cultivar sprayed with FPB in field conditions, where greater injury (13%) has been reported for applications in 5-leaf rice than at 1-leaf (<3%) [36]. These results are more related to the uncontrolled conditions at the field than the size of the rice.

In general, FPB injury increases with dose particularly when applied at the early growth stage and to rice planted earlier. Dawn and dusk application of 2,4-D and dicamba suggest an inverse relationship in the activity and efficacy in *Amaranthus palmeri* S. Watson, a consequence of more translocation and ABA-induced stress [47]. However, further studies in controlled conditions must be done to prove the interaction of FPB, rice, and solar radiation.

For FPB, greater injury recovery has been reported at warmer (32/24 °C, day/nighttime) rather than cooler temperatures (24/17 °C day/nighttime) [35]. In this study, for the temperature averaged over the years and spraying time, five days after spraying was lesser for early planting time (20.6 °C) than medium and late planting time (22.3 °C, 23.9 °C, respectively). This variable may explain the higher injuries and the lesser recovery observed on early application treatments since low temperature may reduce the FPB metabolism [35]. Finally, FPB injury increases with dose but does not affect yield. Therefore, these observations lead us to evaluate the effect of the addition of P450 inhibitors and the expression of candidate genes related to the FPB metabolism and temperature, as shown herein.

Regarding the effect of P450 inhibitors on rice response to FPB, our result is similar to those observed on field experiments carried out in Arkansas rice areas [36]; they reported 2% of rice injuries when they applied FPB (30 and 60 g ai ha$^{-1}$) plus malathion (700 g ai ha$^{-1}$), and there was no detected lasting adverse effects on yield. The label recommendation does not allow the use of malathion on tank-mix or seven days before FPB application [27]. Thus, with both results, this restriction can be changed, and farmers could use malathion safely prior to FPB application or a tank mix. On the other hand, the effect of two P450 inhibitors may have a negative effect on growth and injury, but the doses are still over the commercially recommended; thus, it seems to not compromise the selectivity [27]. Contrary to Wright et al.'s [36] conclusion, which suggests that there is no dependency of P450 on the degradation of this herbicide in rice, we considered that there could be more interaction of FBP with P450 inhibitors in plants since there have been reported 356 CYP genes encoding P450 enzymes on rice [48]. Hence, the addition of two inhibitors, dietholate followed by PBO, may inhibit other CYP isoenzymes that malathion does not do. An example of this is when triazole, imidazole, and pyrimidine derivatives inhibited cytochrome P450 isoenzymes selectively [49]. However, currently, there is no report of the potential accumulative effect of two inhibitors in rice tolerance to FPB. Therefore, more studies must be done in order to consider this effect on rice yield.

Currently, $GR_{50}$ or $ED_{50}$ of FPB has not been reported to rice cultivars. However, the response of cultivars to sequential applications of FPB has been reported [35]. The hybrid CL745 injuries at two FPB sequential doses of 30 g ai ha$^{-1}$ (total 60 g ai ha$^{-1}$) were around 46% and were 64% at 60 g ai ha$^{-1}$ (total 120 g ai ha$^{-1}$) when applications were made four days apart. Thus, a similar result has been reported regarding differential rice cultivars' response to FPB, and the values of the injuries are similar to those observed for BRS Pampeira at 120 g ai ha$^{-1}$. This result supports the hypothesis proposed in this research that cultivars show different levels of tolerance possible by differential metabolism activity or differential bio-activation.

In both cultivars, doses causing $ED_{50}$ and $GR_{50}$ were over the recommended maximum per season (60 g a.i. ha$^{-1}$) and taking into account the label, it must be applied in two separate applications [27]. Therefore, there is no risk on field conditions considering the

maximum doses per season [27]. Further studies could evaluate the metabolism of FPB among cultivars to prove this hypothesis.

Regarding temperature regimes, warmer temperatures tend to exacerbate FPB injury in rice. However, this response varied by the cultivar being examined. A rice tolerance experiment to florpyrauxifen-benzyl rates conducted in a growth chamber at warm and low temperatures (32/24 °C and 27/18 °C day/nighttime) reported that the long-grain pure-line (CL111) was more tolerant over warm temperature conditions than the medium-grain pure-line CL272, with 18% of plant injury at 60 g a.i. ha$^{-1}$, and the long-grain hybrid CLXL745, with 25% of plant injury at 60 g a.i. ha$^{-1}$ [35]. The cultivar IRGA 424 RI used in this experiment corresponds to a medium non-hybrid cultivar; thus, the injury increases in treatments where temperature increased after spraying coupled with what has been reported for medium-grain pure-line CL272 cultivar or long-grain hybrid CLXL745 at 32/34 °C. In Section 3.1, FPB does not show lasting effects on yield. Therefore, the injuries observed in these results do not represent adverse effects on yield.

The differential tolerance of rice cultivars to herbicides has been reported, and this is considered a result of differential crop metabolism rate among cultivars. One example of this has been reported for triclopyr, another synthetic auxin herbicide, where rice cultivars Mars and Tebonnet (15% and 16% of the injury, respectively) were more tolerant than Lemont (25% of injury) [46]. Another common example is the differential tolerance to imazamox among Clearfield® rice cultivars [50]. Hence, it is likely that FPB is metabolized in rice plants, and then temperature changes after treatment affect this metabolism rate, causing the injuries observed in this study. Moreover, the non-bioactivation of the FPB-ester form may be involved since the rice response to temperature affected the activity or expression of enzymes related to this process. Therefore, these observations lead us to evaluate the expression of candidate genes related to herbicide metabolism such as the P450 enzyme (*CYP71A21*) and glutathione S-transferase (*OsGSTL3*) and a possible enzyme related to FPB bio-activation, esterase activity enzyme (*OsWAKL21.2*).

*CYP71A21* is a gene encoding for a P450 monooxygenase enzyme in rice seedlings, described as responsive to herbicides, including auxins [51]. This gene has been reported to be upregulated in *E. colona* resistant to quinclorac (5.1-fold change) compared to de quinclorac-susceptible [52]. Our results demonstrated no upregulation of this gene in rice after FPB application; however, other genes from this family may be affected by FPB application and different environmental factors. For example, *CYP72A1* increases expression in *E. colona* treated with FPB (4.2-fold) compared with the untreated which was down-regulated (−0.3-fold), and this expression was greater induced when plants grow at optimal conditions (30 °C: 4.2-fold) and drought stress (4.2-fold) than at heat stress (45 °C: 3.2-fold) and well-watering (1.7-fold) [53,54]. Correspondingly, *CYP72A15* was greater induced at 45 °C rather than at 30 °C (4.5-fold and 0.1-fold, respectively). Therefore, future studies should focus on evaluating other genes from the *CYP72A* family more than *CYP72A*21 in order to elucidate the interaction between FPB and temperature in rice.

*OsGSTL3* gene encodes glutathione s-transferase, a widely known enzyme that confers herbicide tolerance in rice [55]. GST expression has been described as providing plants the capability to tolerate herbicides. For example, organ-specific expression of *AtGSTU19* confers tolerance of chloroacetamide in *Arabidopsis sp.* [56]. Upregulation of *SbGSTF1* and *SbGSTF2* is induced by fluxofenim, a safener used in seed treatment in *Sorghum halepense* [57]. *OsGSTL2* transgenic rice improves glyphosate and chlorsulfuron tolerance [55]. *OsGSTL3* confers quizalofop-p-ethyl resistance to *Polypogon fugax* [58] Herein, it was observed that there was a differential profile in the expression of *OsGSTL3* as a product of temperature changes over time, probably by its role as ROS scavenger [59]. However, it cannot be proven if the FPB spraying caused an overexpression. Since the increase in FPB rate does not show a consistent increase in expression on *OsGSTL3*.

Our results showed that there is a reduction in GR$_{50}$ of rice when applying FPB with two CYP450 inhibitors (dimethoate and PBO), but not when applying malathion. On the other hand, our candidate gene *CYP72A21* did not show differential response among FPB

rates. These results suggest a certain dependency of CYP450 on rice tolerance but not because of our candidate gene. Therefore, further studies should focus on evaluating other genes of this enzyme family on FPB tolerance to rice.

Wall associate kinases (WAKs) are enzymes localized through the cell wall and membrane, and they are capable of sending signals into the cytoplasm [60]. WAKs have an essential role in physiological prosses such as cell elongation, pollen development, and abiotic and biotic stress tolerance [60]. *OsWAKL21.2* encodes to a WAK, which has a dual function as a receptor or co-receptor of wall damage and a receptor of biotic stress [61]. Although the interaction of this gene with SAHs has not been reported, we hypothesize about its role in FPB bio-activation considering the temperature stress and esterase activity outside the cell. *OsWAKL21.2* overexpression in rice induces tolerance to cell wall degradation products and immune response to lipases/esterases [61]. The *OsWAKL21.2* indicates a downregulation over FPB doses. Thus, this result supports the hypothesis that rice tolerance to FPB may rely on the bio-activation of the ester form and potentially the lack of esterase WAKL enzyme expressions being involved. The lack of esterase activity has already been reported in rice selectivity to cyhalofop-butyl (Acetyl-CoA-carboxylase inhibitors) herbicide [62] and proposed as a resistant mechanism of weeds to auxin herbicides [22]. Additionally, we observed that *OsWAKL21.2* gene expression might vary among high temperatures before or after FPB application 24 HAS (Table 5); thus, the FPB bio-activation may be affected. However, further experiments must be done to confirm the role of *WAKL21.2* in rice sprayed with FPB, additionally to prove the temperature effect on the expression of this gene and the consequence of this interaction in rice selectivity.

## 5. Conclusions

This study contributed to the knowledge about the performance of florpyrauxifen-benzyl (FPB) selectivity to rice in field and controlled conditions. In summary, our research showed that FPB is selective to rice in Brazilian conditions with some highlighted points. As observed at early and medium planting time, injuries in rice can be promoted by low solar radiation and low temperature surrounding the FPB application. Increased FPB doses promote more significant injuries, especially when it was sprayed at $V_2$ and $V_6$. The differential tolerance of rice cultivars to FPB seems to not compromise selectivity in the field. The addition of P450 inhibitors does not compromise the selectivity, and the two candidate genes (*CYP71A21* and *OsGSTL3*) related to the metabolism of herbicides were not upregulated by FPB, whereas a candidate gene related to the bio-activation of FPB-ester (*OsWAKL21.2*) was upregulated. Finally, this study provides farmers and agronomists the confidence to use FPB in Brazilian conditions, increasing the use of new tools in integral weed management. Moreover, this study contributes to expanding the knowledge of auxin herbicides and hints about the rice selectivity mechanism to FPB, especially regarding bio-activation.

**Supplementary Materials:** The following are available online at https://www.mdpi.com/article/10.3390/agriculture11121270/s1, Figure S1: Daily mean, maximum and minimum air temperature, rainfall (gray shading), and solar radiation throughout the experiment. The continuous line represents planting date and discontinue line represents spraying dates. Planting time abbreviations E: Early planting time (blue), M: Medium planting time (red), and L: Late planting time (orange). Spraying time abbreviations V2: collar formation on leaf two on the main stem; V6: collar formation on leaf six on the main stem, and R0: Panicle development has initiated. (Source: Embrapa Clima Temperado weather station). Table S1: Temperature regime "Factor B" of growth chamber experiment. Table S2: Oligonucleotides used in this study for RT-qPCR assay. Table S3: Yield components and analysis of deviance type II Wald chi-square test of rice as affected by planting time (two seasons 2019/2020; 30 September, 25 October and 11 November; and 2020/2021; and 25 September, 20 October and 10 November), spraying time ($V_2$, $V_6$, and $R_0$) and florpyrauxifen-benzyl rates (0, 30 and 60 g ai ha$^{-1}$). Table S4: Parameters estimate of the dose response curve of rice plant injury evaluated at three, seven, 14, 21 and 28 days after florpyrauxifen-benzyl treatment as affected by P450 inhibitors applied one

hour before for malathion and Piperonyl butoxide, and at seed treatment for dietholate. Table S5: Best Linear Unbiased Predictor BLUP's of runs for rice injury for growth chamber experiment.

**Author Contributions:** Conceptualization, J.C.V., L.A.d.A. and G.P.; methodology, A.D.C.B., J.C.V., A.A., V.E.V. and L.A.d.A.; software, J.C.V.; validation, G.P., L.A.d.A., A.A., A.D.C.B. and E.R.C.; formal analysis, J.C.V.; investigation, J.C.V.; data curation, G.P., V.H., V.E.V., L.A.d.A., A.A., A.D.C.B. and E.R.C.; writing original draft preparation, J.C.V., L.A.d.A. and G.P.; writing review and editing, J.C.V., V.H., G.P., V.E.V. and L.A.d.A.; visualization, J.C.V.; supervision, L.A.d.A., A.A., A.D.C.B. and E.R.C.; project administration, L.A.d.A., A.A., A.D.C.B. and E.R.C.; funding acquisition, L.A.d.A. and E.R.C. All authors have read and agreed to the published version of the manuscript.

**Funding:** This research received external funding from CNPq (Conselho Nacional de Desenvolvimento Científico e Tecnológico) for the Research Fellowship of L.A.A./N.Proc. 310830/2019-2. This study was financed in part by the Coordenação de Aperfeiçoamento de Pessoal de Nível Superior Brasil (CAPES)-Finance Code 01 by providing the student assistantship of J.C.V.R. And part of the experiment was financed by Corteva Agriscience.

**Institutional Review Board Statement:** Not applicable.

**Informed Consent Statement:** Not applicable.

**Data Availability Statement:** Not applicable.

**Acknowledgments:** The authors thank the Federal University of Pelotas for the staff support and all the external revisor for their critical contribution on drafts of this paper.

**Conflicts of Interest:** A.D.C.B. declare that she is employee of Corteva Agriscience but this did not affect her ability to contribute to this manuscript. The rest of authors declare no conflict of interest.

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
