# Peer review of "Florpyrauxifen-Benzyl Selectivity to Rice"

_agriculture, doi:10.3390/agriculture11121270_

Round 1
Reviewer 1 Report
Αbstract: Information on experimental design and herbicide rates should be included in the abstract. Expressions such as Lastly should be removed. Split some sentences into smaller ones. Some sentences need to be shortened. I have also an additional comment. Can you provide a sentence about further research objectives that should be addressed?
- Introduction
Lines 35-43: Is weed competition a major obstacle in rice productivity? Prove it by including more references about yield losses recorded recently in relative case studies. The same applies for herbicide resistance. Some English editing is required.
Lines 51 and 67: Would like to see information about populations that are resistant to other herbicides than quinclorac because this active ingredient is not approved in Europe. Talk about herbicides that are used in all over the world where rice is grown. Alternative herbicides are also effective on glyphosate-resistant populations belonging to Echinochloa spp. genus (check a study by Travlos et al. 2020 conducted on E. colona).
Line 74: Acetyl-CoA-carboxylase
- Materials and Methods
In general, I suggest divide Subsection 2.1 to Sub-Subsections 2.1.1 (site and plant material description), 2.1.2 (experimental procedures), 2.1.3 (evaluations), and 2.1.4 (statistical analysis).
Line 135: Information about soil texture and physical properties in the upper layer (0-30 cm) are missing. It is very important to know about soil pH, organic matter and nitrogen content etc. Moreover, I would like to see some key traits of your selected plant material (e.g. cultivar). Site description should also include coordinates of the exact experimental location (and also elevation).
Lines 136-137: kg “seed” ha-1. Row spacing is missing. I also see no information about seedbed preparation (ploughing, harrowing timing and depth etc.)? How was flooding was carried out? Trade names and companies’ names of the fertilizers used? How much amount of water was needed? What was water height in the flooded rice field?
Line 138: “resulting in the addition of”
Line 140: two-factorial.
Line 141: The text in the brackets is confusing.
Lines 147: Herbicide manufacturer’s name is missing. Include trade names and manufacturer’s names also in lines 153, 156.
Lines 146-150: Long sentence.
Lines 150-151: What about borders between adjacent plots?
Line 162: Trade names and manufacturer names are missing for the spraying equipment.
Lines 167-170: Doubt if visual ratings are reliable. I leave the decision to the Editor’s choice. I would prefer to see spectrometer (NDVI for example) and destructive measurements (weed dry weight for example) that should also be correlated.
Lines 169-173: First, what did you mean in a linear meter? Did not you measure 1000 grain weight? It is a major component of yield. Moreover, the test needs some editing to be more accurate and less confusing.
Line 175: Please include the reference of Shapiro and Wilk. It is available on Google Scholar. What about data homoscedadicity? Did you perform also Levene’s test? In addition, how were the means separated?
Line 186? Why to consider planting date as a random factor? Please explain.
Line 186? Can you prove with a table that data could be actually pooled over growing seasons? Did you include year’s effects on your analysis as fixed effects? Moreover, you should include a table with climatic data since it is a major factor influencing the outcome of any field trial (and a major component of site description). You should also explain weather’s effect on your results. Was that the reason why year’s effects were not significant (giving you the option to present combined data across seasons)?
Line 207: Why were the number of replications different?
Line 210: “The factor B consisted of three different levels “
Lines 217-218: More details about day and night temperature, RH, and photoperiod are required.
Line 219: Herbicides were applied at the 3-leaf growth stage of rice.
Line 291: Mention the cutting height.
- Results
First of all, I don’t understand what you mean in Lines 329-330. For example, I would prefer to see ”the interaction between… affected rice injury” Include also (P-Value <0.001) in brackets. Moreover, I do not see the p-value obtained for each factor. I see only the interactions. This also applies for Table 1.
In this Table 1, crop injury data are not easy to interpret without a lot of effort. I would suggest to prepare a table only for ANOVAs and prepare a separate figure with the injury values (for each factor). It will allow for rapid visual comparison between the different levels of each factor. Include also the standard errors of each of your measurements with vertical bars. Another comment, please include LSD od Tukey HSD values that helped you separate means.
Another comment: How can you compare planting dates if you considered them random effects? All comparisons must be the outcome of statistical analysis and LSD/HSD values should be given. Comparisons (e.g. significant differences) between the different levels of a factor can also be supported from percentages to make the text less wordy and more attractive.
In Figure 1, the horizontal axon is weird. There is a group of numbers next to each other. Prepare the figure again in different a way. Some space is needed between herbicide rates. Would also like to see the standard error of each measurement. Similar are the comments for Figure 2.
Wordy text in lines 470-475: Split in smaller sentences.
Lines 450, 468, 480: Exclude references from this section. The “Results” section is targeted to just focused on what actually happened in your research.
- Discussion
A lot of data are presented here. You should not do that. You have presented all your results in section 3. Thus, you should exclude supplementary figures and place them in section 3. “Discussion” is the section to explain why phenomena occurred and not introduce new points. Moreover, since the manuscript presents multiple multi-factor experiments, I think that 10-15 more references should be cited. In a total of 50 cited references, only five references are introduced in the discussion section.
- Conclusions
Provide us a real conclusion in just one short paragraph of 100-150 maximum words and emphasize what is the "take home" message from this research. Provide us also information about knowledge gaps and future challenges in this research topic.
Reviewer 2 Report
General: This is an interesting piece of research that examined the selectivity mechanism of a new auxinic herbicide in rice and associated weeds. Overall, good work. English usage needs to be improved.
Specific:
L18: Floropyraauxifen-benzyl (FPB) is a new herbicide from a recently introduced chemical class of auxinic herbicides...
L20: Field experiments carried out...
L23: smaller should be younger here
L26: "...inhibitors decreased the dose to cause 50%..."
L51: "...even in those species with confirmed resistance to other auxin..."
L52: "...FBP has the potential to provide efficient..."
L65: "...receptor over other SCFTIR/AFB receptors. For example, Arabidopsis thaliana L. mutants with a a silenced SCFTIR/AFB5 were susceptible to 2,4-D, but not to FBP ."
L67: "...novel site of of action allows FPB..."
L70: product yield should be crop yield
L78: What does 76%, 21%, and 3% refer to here?
L92: "The selectivity mechanism of FPB in rice has not been fully described..."
L103: due to the recent introduction
L106: experiments
L119: Considering the recent introduction of FPB...
L154: What does complementary burndown mean? Was this a separate treatment where glyphosate was applied with the PRE herbicide?
L157: For the 2019/2020 season, imazapic + imazapyr (24.5 and 73.5 g ai ha-1) was applied postemergence to keep plots free of weeds.
L184: analysis of variance; should be there were no significant year by treatment interactions
L187: What is aggrupation?
L190: "...instead of linear models..."
L238: data were pooled
L333: What is aggrupation?
Table 1: This is a clunky and inefficient way to analyze this data. This is a structured data set with continuous variable of spraying time, herbicide rate and planting time. It is being analyzed as though these variables were discreet using a mean's separation test. This clunky analysis makes for a clunky discussion. A response type analysis would provide a more meaningful discussion.
L513: In general, FPB injury increases with dose particularly when applied at early growth stage and to rice planted earlier.
L529: rather than cooler temperatures..."
L534: Finally, FPB injury increases with dose but does not affect yield.
L550: This sentence makes no sense.
L562: Warmer temperatures tend to exacerbate FPB injury in rice. However, this response varied by the cultivar being examined.
Reviewer 3 Report
Paper is related to the relatively new florpyrauxifen benzyl herbicide. I have some concerns to address in the manuscript, properly response to those is required.
-L25. Are the authors saying that malathion does not affect selectivity but addition of two P450 inhibitors have influence on the values obtained, is there any explanation about this?
L102-103 Please provide examples and references.
L147 Please change to non treated/ non-treated through ms and avoid untreated.
L153 re-write sentence to avoid glyphosate twice
L154-155 Authors need to explain those lines since it would have an impact on the results obtained, other studies have demonstrated that glyphosate affects yield when applied even at far much lower rates than the used in this manuscript (see RICE (Oryza sativa) RESPONSE TO LOW GLYPHOSATE RATES AS INFLUENCED BY CULTIVAR, GROWTH STAGE, AND IMAZETHAPYR APPLICATIONS )
L200-201. This kind of arrangement have a great impact on the development of rice plants, the best thing is to have planted a single plant per pot to avoid competence among plants. Thus, results could be different to those obtained in the manuscript.
Paragraph 205-218. I am lost in terms of these studies, why authors performed these experiments if FPB is already used in Brazil on certain varieties, only two?
L204 modify sentence to avoid "recently" or be precise.
L230, authors mean statistic environment?
L244-249 See comment above regarding the number of plants per pot, in this case competence and stress among the plants could affect results obtained.
L275-275 Please merge this section with the previous one since most of the information provided here was already stated before, consider a specific section for analysis.
L278. So authors in this case used four commercial rice seeds per pot? then why authors did not include the other variety to have consistency through the paper?
L301 please modify sentence to avoid misunderstanding with initial material used.
RNA section and cDNA synthesis.
Is not clear why authors decided to choose those genes, one P450 when there are several, but why to choose a GST if initial experiments did not included a GST inhibitor and in any case the same issue as before, there are several GSTs. This comment affect the objectives section.
Table 1 is not understandable at all, consider in changing to sup material, at 42 DAA how it can be different medium 0.0 in R0, 60 from R0 60 at early planting?
L450 so authors mean that this gene is responsible for FPB tolerance?
L458. since there were differences between cultivars in the previous section why authors did not include pampeira in this assay?
Table 5.
How authors explain the upregulation of the gst at 60 6HAS is it one single gene that is conferring tolerance in rice? Furthermore, an full explanation of the downregulation of the main gene p450 is needed when it was demonstrated before there were differences in the p450 inhibitors.
Round 2
Reviewer 1 Report
Important clarifications were added. The quality of this MS is now improved.
Author Response
Thank Dr.
Reviewer 3 Report
Manuscript has been reviewed again, authors have addressed some questions, however other remained unanswered or a weak response was provided.
-L25 Reply. Author's failed to provide an adequate reply. Just cited another source and tried to justified with more experiments.
-L154-155. Despite that "minimal" contact with glyphosate, seedlings would be dead, even more when sprayed at field rates.
-L278 comment. Is still unclear why authors did not include all rice varieties in the experiments.
-P450 and GST selection genes, is unclear why authors chose those genes, they just refer to another source, since FPB can affect a diverse group of genes including TIR and AFB genes, so why to select those two? There are several GSTs and P450s that would be "responsive" in rice, including auxins meaning that, as expected be over/under expressed under external/internal stimuli. Regarding the OsGSTL3 are authors saying that this is a target gene for herbicide discovery? Apart from that I have revised reference provided (48) and it does not even mention this gene.
This comment has not been solved:
Table 1 is not understandable at all, consider in changing to sup material, at 42 DAA how it can be different medium 0.0 in R0, 60 from R0 60 at early planting?
L458. Authors failed to provided an appropriate answer for this question.
Response given to last comment, is corroborating that the selection of the genes used in this study was inadequate.
